# Bacterial Infections in Intensive Care Units: Epidemiological and Microbiological Aspects

**DOI:** 10.3390/antibiotics13030238

**Published:** 2024-03-05

**Authors:** Maddalena Calvo, Stefania Stefani, Giuseppe Migliorisi

**Affiliations:** 1U.O.C. Laboratory Analysis Unit, A.O.U. “Policlinico-San Marco”, Via S. Sofia 78, 95123 Catania, Italy; maddalenacalvo@gmail.com (M.C.); stefania.stefani@unict.it (S.S.); 2Department of Biomedical and Biotechnological Sciences (BIOMETEC), University of Catania, 95123 Catania, Italy; 3U.O.C. Laboratory Analysis Unit, A.O. “G.F. Ingrassia”, Corso Calatafimi 1002, 90131 Palermo, Italy

**Keywords:** ICU, microbiology, antimicrobial stewardship, diagnostic workflow

## Abstract

Intensive care units constitute a critical setting for the management of infections. The patients’ fragilities and spread of multidrug-resistant microorganisms lead to relevant difficulties in the patients’ care. Recent epidemiological surveys documented the Gram-negative bacteria supremacy among intensive care unit (ICU) infection aetiologies, accounting for numerous multidrug-resistant isolates. Regarding this specific setting, clinical microbiology support holds a crucial role in the definition of diagnostic algorithms. Eventually, the complete patient evaluation requires integrating local epidemiological knowledge into the best practice and the standardization of antimicrobial stewardship programs. Clinical laboratories usually receive respiratory tract and blood samples from ICU patients, which express a significant predisposition to severe infections. Therefore, conventional or rapid diagnostic workflows should be modified depending on patients’ urgency and preliminary colonization data. Additionally, it is essential to complete each microbiological report with rapid phenotypic minimum inhibitory concentration (MIC) values and information about resistance markers. Microbiologists also help in the eventual integration of ultimate genome analysis techniques into complicated diagnostic workflows. Herein, we want to emphasize the role of the microbiologist in the decisional process of critical patient management.

## 1. Introduction

The intensive care units (ICUs) contain a complex variety of microbial communities whose biodiversity and clinical implications are not sufficiently comprehended. Critical hospital wards constitute alarming indoor settings, where microorganisms can easily contaminate surfaces, devices, and healthcare personnel. This condition contributes to reaching a specific setting-related ecosystem due to extensive sanitation protocols, prolonged antimicrobial treatments, and protracted recoveries. Each aspect may select species that can subvert the native microbial communities, developing hypervirulence, resistance patterns, and biofilm formation capability [1,2].

The intrinsic ICU-related ecosystem provides the ideal substrate to generate an unsettling scenario, comprising the persistence of the microorganisms, the difficult-to-treat appearance of the pathogens, and the intense patients’ vulnerability to infections [2]. ICU patients frequently have comorbidities, immunosuppression, and seniority and often experience invasive procedures and external device implantation. On the one hand, these risk factors establish a critical ICU patient’s susceptibility to colonization and possible hospital-acquired infections (HAIs). On the other hand, ICU-related pathogens may express all their virulence or resistance features in these fragile patients.

Specifically, multidrug-resistant (MDR) pathogens hold high morbidity and mortality rates among ICUs [2]. On that premise, it is essential to correctly distinguish between colonization and infection conditions by applying standardized clinical and diagnostic protocols. Clinical microbiology support holds a crucial role in the diagnostic algorithms’ definition to rapidly reach data on identification and antimicrobial susceptibility testing. Eventually, the complete patient evaluation requires integrating local epidemiological knowledge into the best practice and the standardization of antimicrobial stewardship programs.

## 2. Surveillance and Diagnostic Aspects

The prompt identification of an infection process among ICU patients involves a multiple evaluation. Foremost, it is essential to estimate clinical signs such as pyrexia and tachycardia, which are not specific in this healthcare setting. As a consequence, the clinical approach needs to perform laboratory parameters such as white blood cell count (WBC), C-reactive protein (CRP), procalcitonin (PCT), presepsin, and pro-adrenomedullin. The alteration of these biomarkers allows for the consolidation of the suspected infection, while their normalization may suggest antimicrobial treatment de-escalation. Moreover, biomarkers often show great specificity rates in comparison or in combination to sequential organ failure assessment (SOFA) and other clinical scores [3,4,5]. Clinical and laboratory information describes the general condition of the patients, representing the first step to correctly defining a diagnostic microbiological workflow. Clinical microbiologists must help clinicians in distinguishing colonization from infectious conditions. For this purpose, early information about microbiological data results in preventing fatal outcomes and concerning outbreaks. Moreover, microbiological information guides patient management, co-ordinating organization, and infection control procedures.

The ICU settings may involve diversified conditions such as hospital-acquired or community-acquired infections (CAIs), which require clinical microbiologist recognition and reading [6,7]. The most valuable tool in recognizing colonization or infection conditions is the correct planning of surveillance programs. Surveillance may be accomplished through passive protocols, periodically gathering retrospective laboratory data about specific hospital units. On the one hand, this epidemiological research is easy to perform over extended time intervals and does not involve high costs. Furthermore, it collects complete knowledge about isolates, resistance profiles, and effective therapeutical eradications. Therefore, clinicians can manage patients’ conditions, specific aetiology suspicion, and empirical treatment choices by combining single-centre and multi-centre surveillance [6,8]. All this information contributes to settling pragmatic infection control strategies, helping to define active surveillance protocols. The active screening policies strictly depend on the local epidemiology and the features of the most critical hospital units. Patients at high risk of infection or cross-transmission, such as ICU patients, represent the most reliable candidates to apply for an active surveillance program.

The active surveillance aims to implement standard precautional measures (admission screening), rapidly prevent pathogen transmission within the unit during the recovery (repeated screening), correctly manage possible inter-ward transfers (discharge screening), and finally screen vulnerable patients before invasive procedure (pre-operative screening). Figure 1 illustrates some indications for sampling timing.

The mentioned screening strategies should be maintained for long-term time intervals within the hospital setting to know the patients’ status. The screening tests can utilize culture-based methods or rapid molecular techniques. A negative result does not exclude the presence of an MDR microorganism below the detection limits of each diagnostic procedure. Positive results should be rapidly provided to clinicians and centre direction to manage eventual contact precautions and enrich epidemiological databases [9]. The gathered information integrates national and international surveillance data about MDR microorganisms into critical healthcare settings [10,11].

## 3. Epidemiological Details

Several epidemiological surveys documented the Gram-negative bacteria supremacy among ICU infection aetiologies. Specifically, *Klebsiella* species hold the primate, but *Klebsiella pneumoniae* detains the highest percentages. Furthermore, *Escherichia coli*, *Pseudomonas aeruginosa*, and *Acinetobacter baumannii* could similarly and severely impact ICUs. Among Gram-positive bacteria, *Staphylococcus aureus* and *Enterococcus* species prevail [12,13,14]. Despite this conception, there is no specific and recognized prevalence in mortality and unfavourable outcomes concerning the isolated species. The common risk factors, such as older age or comorbidities, are the basic features for the death rate increase in ICU patients [12,13].

Therefore, a specific bacterial aetiology does not significantly affect the negative outcome rate, and this assumption generally interests the ICUs worldwide.

On the other hand, geographical areas expose country-specific variations in lifestyle, healthcare facilities and accesses, nutritional state, and vaccine or antimicrobial drug availability. Each aspect has a relevant impact on the ICU patient status, even if no data may establish an actual prevalence among all of them [12,13,14,15]. The mentioned premise strictly assumes antimicrobial susceptible microorganisms. Unfortunately, ICUs often notice MDR aetiologies, which dramatically impact patients’ outcomes. The Gram-negative resistance episodes mainly regard third-generation cephalosporins (33.3%) and carbapenems (20.4%) among Enterobacterales. The carbapenem-resistant isolates also belong to *P. aeruginosa* (30.2%) and *A. baumannii* (77.7%) [16,17,18]. Moreover, literary data confirm methicillin-resistant *S. aureus* (MRSA) and vancomycin-resistant *Enterococcus* (VRE) as potential ICU mortality causes despite their low prevalence [12,19,20,21].

### 3.1. Acinetobacter baumannii

*Acinetobacter baumannii* usually exhibits a lower virulence spectrum, although it has the ability to adhere to surfaces, medical devices, and personnel hands. Moreover, the same species frequently colonizes the patients’ oropharyngeal, cutaneous or gastro-intestinal districts within 48 h of ICU recoveries [22,23]. The opportunity for *A. baumannii* to develop a biofilm formation increases the chance of resisting the nosocomial environment. Specifically, biofilm-related genes, such as ompA, bap, and blaPER-1, help the microorganism to maintain its persistence [23]. Such bacterial quality largely contributes to the sticking of ICU mechanical ventilators required for acute respiratory failure management among recovered patients. Although these devices improve morbidity rates of the patients, they also correlate with *A. baumannii* ventilator-associated pneumoniae (VAP) episodes. The COVID-19 pandemic even aggravated *A. baumannii*-related VAP, extensively contributing to mechanical ventilation requests within ICUs [24,25]. Severe respiratory conditions frequently precede hematic dissemination and systemic infections, complicating therapeutical management due to *A. baumannii* resistance patterns. For instance, carbapenem-resistance *A. baumannii* (CRAB) predominates within ICUs, mostly due to blaOXA-23, blaOXA-66, and blaOXA-51 genes [21]. The mentioned resistance markers integrate an extended resistance profile, leading to the *A. baumannii* insertion into the alert microorganisms’ lists [26].

### 3.2. Klebsiella pneumoniae

*K. pneumoniae* accounts for high healthcare-associated infection (HAI) percentages, especially among critically ill patients. The species usually integrates human gut microbiota, even if it can colonize the upper respiratory tract, the hospital environments or the staff. At admission to the ICUs, patients with *K. pneumoniae* show a concerning chance of developing severe infection due to the same microorganism [27]. This species usually causes acute infectious conditions due to a large virulence spectrum, mainly related to specific plasmid-associated genes. For example, the gene loci iro for salmochelin biosynthesis, and the gene loci iuc for aerobactin synthesis largely contribute to increased siderophore production to face iron-limiting living conditions [28,29].

*K. pneumoniae* strains present the polysaccharide capsules to evade the patient’s immune response and the lipopolysaccharides to exacerbate the virulence expression against the human host. Furthermore, the rmpA gene regulates the mucoid phenotype, leading to a potential hypermucoviscous *K. pneumoniae* strain, whose increased capsule production enhances the virulence expression. Additionally, *K. pneumoniae* can potentially use fimbriae and produce biofilm-related adhesins, increasing at the same time its capability to disseminate or colonize anatomic districts and surfaces [29,30].

*K. pneumoniae* is also distinguished for complicated susceptibility profiles. Carbapenem resistance due to β-lactamases production and/or membrane permeability alterations is the most relevant phenomenon to notice about this species. Carbapenem-resistant *K. pneumoniae* (CR-Kp) mainly expresses its pattern through *K. pneumoniae* carbapenemases (KPC), OXA-48-like enzymes, and metallo-β-lactamases (MBLs), such as Verone-imipenemase (VIM), imipenemase (IMP), and New Delhi metallo-β-lactamases (NDM) [31,32,33]. According to these markers, *K. pneumoniae* enters the worldwide alert microorganisms’ monitoring programs [10]. Surveillance samples, such as rectal and oropharyngeal swabs, allow for promptly identifying potential MDR strains’ colonization, preventing severe infection spread.

A substantial amount of literary data documented how dissemination episodes strictly depend on previous MDR *K. pneumoniae* rectal colonization [34,35,36,37]. *K. pneumoniae* respiratory colonization may involve mechanical ventilation in the VAP generation and the potential subsequent dissemination [38,39]. Several studies highlighted how ICU *K. pneumoniae* isolates correlate with high-risk clones (HiRiCs), whose features list colonization abilities, epidemicity, nosocomial and/or host persistence, and antimicrobial resistance due to specialized genetic populations [40,41,42].

### 3.3. Pseudomonas aeruginosa

*P. aeruginosa* represents the most common aetiological agents among ICU respiratory infections. Its virulence spectrum is the most relevant marker for this species. First, *P. aeruginosa* has several surface appendages, such as type IV pili and flagella, which allow a promising movement and adhesion capability. Furthermore, outer membrane structures such as lipopolysaccharide and secretion systems (T1SS-T6SS) stimulate the host immune response and produce toxins or biofilm-related compounds such as alginate. The biofilm formation leads to an essential environmental persistence, although with difficult growth conditions. Additionally, the quorum sensing phenomenon modulates biofilm expansion and virulence factor production, regulating bacterial cell-to-cell interactions [43]. Both biofilm and pili potentiate *P. aeruginosa* capability to adhere to living substrates and medical devices. The quorum sensing regulation leads to regular and specific polysaccharidic bacterial structures, providing an ideal growing environment for *P. aeruginosa* cells.

Critical patients often require mechanical ventilation to manage acute respiratory failures, and this need may correlate with *P. aeruginosa* lower respiratory tract infections due to a frequent previous microorganism mechanical ventilator or pharyngeal tract colonization. According to this, surveillance programs should integrate respiratory sample monitoring to document *P. aeruginosa* colonization, whose subsequent infection increases mortality rates in fragile patients [43,44]. Moreover, potential resistance patterns complicate *P. aeruginosa* respiratory infections among ICU patients. Specifically, MBLs or KPC production may lead to carbapenem resistance, along with the most common membrane permeability alterations related to outer membrane protein (Omp) depression or efflux pump overexpression [45,46]. Carbapenem-resistant *P. aeruginosa* joins nosocomial alert microorganisms within the European Centre for Disease Prevention and Control (ECDC) epidemiological reports [10].

### 3.4. Escherichia coli

*Escherichia coli* normally integrates into the human gastrointestinal microbiota, as well as *K. pneumoniae*. Unfortunately, it also represents a facultative pathogen which may cause urinary or systemic infection, disseminating from the native gastrointestinal district. The extra-intestinal migration correlates with virulence factor expression. Numerous compounds as adhesins, toxins, iron acquisition factors, lipopolysaccharides, polysaccharide capsules, and invasins enhance *E. coli* capability to invade blood and tissues, expressing adherence and persistence [47]. Critical patients, such as immunocompromised hosts, are more predisposed to *E. coli* dissemination. Therefore, this microorganism frequently causes bacteriemia and sepsis among ICU patients. *E. coli* may carry extended-spectrum β-lactamases (ESBL) genes (bla genes), which take place within mobile elements (plasmids) or chromosomal positions. The blaCTX-M genes are the most clinically relevant resistance markers in *E. coli*, which occasionally carry carbapenem-resistance genes such as blaNDM and blaKPC [48]. Resistant *E. coli* is included within the ECDC alert microorganisms’ lists, especially due to its capability to transfer resistance genes [10].

### 3.5. Staphylococcus aureus

*S. aureus* frequently invades nosocomial settings due to its capability to adhere to different surfaces. This species easily forms biofilm on indwelling medical devices, producing virulence factors to evade the host’s immune response. Specifically, *S. aureus* produces haemolysins, leucocidins, proteases, enterotoxins, and exfoliative toxins as relevant virulence factors. A regulatory system, the agr system, is a quorum sensing system involved in these factors’ production along with sar, rot, and mgr transcriptional regulators [49]. *S. aureus* toxins damage host cell membranes, causing cytolytic injuries and cellular function alterations. The toxin production emerges as a relevant feature for severe infectious diseases among ICU patients [50]. Although medical devices often represent the staphylococcal source, fragile patients are frequently prone to opportunistic *S. aureus* infection. The species may colonize the upper respiratory tract, causing a subsequent lung infection after viral infections or severe immunosuppression. The same conditions may contribute to systemic infections, such as *S. aureus* bacteriemia and sepsis [51,52].

Resistance patterns complicate *S. aureus* infection treatment. The methicillin-susceptible *S. aureus* (MSSA) and the methicillin-resistant *S. aureus* (MRSA) share the same virulence factor spectrum, supporting the importance of promptly identifying resistance markers to really apply an effective therapy [50]. The ECDC surveillance programs included MRSA within alert microorganisms list [10]. *S. aureus* nasal, pharyngeal, and gut colonization are extremely common in community subjects, but persistent carriers seem to contrast potential *S. aureus* infections through anti-staphylococcal antibodies. Despite this assumption, the carriers represent a harmful staphylococcal source, especially when they typically attend healthcare settings. All the critical hospital settings need to adequate their infection control protocols to the *S. aureus* research on rectal and respiratory surveillance samples from potential asymptomatic colonized patients [53,54,55,56,57]. Certainly, epidemiological practices such as hand sanitization and patient isolation support S. aureus eradication in ICUs [58,59]. Antibiotic treatment is still the gold standard for front staphylococcal infection in intensive care patients, especially when MRSA is confirmed [60].

### 3.6. Enterococcus Species

*Enterococcal* species account for about 6.1–17.5% of isolated strains from European recovered patients, even if these species initially integrate into the human gut microbiota. These percentages are mainly related to *Enterococcus faecalis* and *Enterococcus faecium*. They frequently cause urinary tract infections, systemic infections, endocarditis, or wound infections after surgical intervention. Furthermore, enterococci may colonize medical devices, causing catheter-related infections [61]. In regard to the virulence spectrum, *Enterococcus* species produces cytolysins, gelatinases, pore-forming toxins, and aggregation substances, which promote tissue invasion, immune evasion, or cellular damage [62]. Vancomycin-resistant enterococci (VRE) currently represent a fundamental healthcare concern due to an increased mortality rate among ICU patients [63,64,65,66]. Surveillance programs include VRE microbiological research on rectal swabs to prevent invasive diseases according to the ECDC protocols [10].

### 3.7. Streptococcus pneumoniae

The community-acquired acute bacterial meningitis (CABM) often requires ICU admission due to severe outcomes and high mortality rates. *Streptococcus pneumoniae* is the most isolated aetiological agent in the case of this condition, which remains fatal in a moderate case percentage (15–33%) despite the diffusion of effective vaccination campaigns [67,68,69,70,71]. Although the possible ICU patient enters with *S. pneumoniae* meningitis, some patients reach the same units for different acute conditions, developing *S. pneumoniae* meningitis, bacteriemia, or pneumoniae after an upper tract colonization (nasopharynx) by the same species. The *S. pneumoniae* capability to cause severe infection depends on the host’s immune conditions and the bacterial virulence pattern [72]. The microorganisms produce a polysaccharide capsule, surface proteins, and the toxin pneumolysin. All these factors contribute to host invasion and damage, but the polysaccharide capsule probably represents the essential pneumococcal virulence feature.

The capsule protects *S. pneumoniae* from phagocytosis, mucus mechanical removal, and antibiotic exposure [73,74]. Some ECDC periodical reports document the importance of monitoring *S. pneumoniae* colonization through surveillance programs, considering the dissemination risk [75]. According to the huge aetiological agents’ spectrum, ICU surveillance choices should depend on the suspected microorganisms to detect. Therefore, a reasonable approach may involve multiple sites to sample. Table 1 describes a suggested sampling strategy within ICU settings.

### 3.8. Clostridioides difficile

ICU patients occasionally suffer from *Clostridioides difficile* infections. Specifically, the *C. difficile* infections prevalence among ICU patients accounts for 0.4–4%, slightly impacting on morbidity and mortality rates. An initial colonization (10–20% of ICU patients) may become an infection episode due to patients’ fragilities. Risk factors such as immunosuppression or prolonged broad-spectrum antibiotic treatment should suggest a colonization investigation on stool samples. Culture-based methods, chemiluminescence, or real-time PCR assays may be used to detect *C. difficile* colonies, antigens or nucleic acid. Molecular methods also allow us to determine the *C. difficile* ribotype: some hypervirulent strains, such as the O27, severely impact on critically ill patients due to the hyperproduction of toxins [76].

## 4. Microbiological Diagnostic Procedures

According to several literary data, ICU patients mainly suffer from severe respiratory infections (hospital-acquired infections such as VAP and complicated community-acquired infections also called “CAP”). Urinary tract and bloodstream infections usually reach lower rates in the same patients’ setting. On that premise, the diagnostic microbiology laboratory usually receives respiratory samples from upper or lower tracts, urine, and blood [77].

### 4.1. Conventional Diagnostic Methods

Regarding respiratory diagnostic procedures, the microscopic analysis still represents an essential starting point. Microscopic results may clarify qualitative information about eventual microorganisms, along with cellular elements evocative of inflammatory processes such as white blood cells (granulocytes) and mucus. Furthermore, the Gram stain verifies respiratory samples’ suitability to subsequent diagnostic phases, excluding invalid specimens according to Bartlett’s criteria. Epithelial squamous cells suggest possible sample contamination through the upper respiratory tract, while granulocytes come out on the side of an acute inflammatory condition. After microscopic validation, respiratory samples such as sputum or broncho-aspirate undergo culture exams on enriched and selective agar media. Lower respiratory tract samples such as bronchoalveolar fluid do not need this validation because their anatomic source is clearer than the upper respiratory tract [78]. Blood samples face a variable incubation period within enriched liquid media, which enter an automated incubator. After being flagged as positive, an extemporary Gram stain is prepared. This investigation may furnish preliminary information about microorganisms’ morphology and type, and early orienting antimicrobial treatments. Consequently, the positive blood samples sustain culture assays through enriched agar media [79].

Urinary samples mainly go through culture exams on enriched and selective agar media, along with a preliminary physical and chemical examination, which support eventual evidence of nitrates, leucocytes, turbidity, or pH significant alterations. With regard to incubation conditions, all the culture plates usually require 18–24 h, which may become 48–72 in the case of fastidious microorganism identification [80].

After the required incubation period, features such as microbial quantity, purity, and morphology are analysed. Specifically, the microbial count is essential to certify microbial aetiology and an ongoing infection process at urinary and respiratory sites. Only significant microbial counts match with the suspicion of an infection and its related clinical conditions. Otherwise, blood samples are processed independently from microbial quantification due to their sterile anatomic source. All the polymicrobial results need to be evaluated as possible contaminations according to the related clinical evidence [81]. Significant microbial growths sustain antimicrobial susceptibility testing, which is usually performed through automated turbidimetric and colourimetric technologies. Its result may require a supplementary 24 h interval, accounting for about 72 h of definitive turn-around time (TAT). Polymicrobial growth, fastidious microorganisms, and extended resistance patterns may prolong this interval. The susceptibility profile evaluation provides the minimum inhibitory concentration (MIC) values, which is crucial in planning patients’ targeted therapies [82].

### 4.2. Rapid Diagnostic Methods

On the one hand, the conventional methods are reliable, but they require prolonged processing periods. On the other hand, the recent technological advance proposes several identification and susceptibility testing rapid techniques for a significant TAT reduction. Regarding the identification processes, the matrix-assisted laser desorption/ionization-time of flight (MALDI-TOF) and magnetic resonance methods became predominant in the case of positive blood samples [83,84,85]. Additionally, either the nested-PCR or the real-time multiplex PCR is usually integrated into syndromic panels to rapidly identify microbial nucleic acids directly from respiratory specimens, cerebrospinal fluid, and blood within 45–90 min. These molecular technologies often have a restricted identification spectrum but furnish preliminary information about specific resistance markers [86,87,88].

One of the advantages of using the respiratory syndromic panel is the chance to provide an accessory quantitative information about the hypothesized microbial count. Specifically, the molecular technology furnishes a microbial quantification as genomic copies/mL (bin), which does not have the same relevance as a culture microbial count due to the possible presence of non-vital genomic fragments. Quantitative molecular information should integrate culture microbial count to validate a microbiological diagnosis [89,90,91]. ICU patients often require fast susceptibility diagnostic protocols due to possible systemic infections. Therefore, the European Committee for Antimicrobial Susceptibility Testing (EUCAST) proposed a rapid antibiotic susceptibility test (RAST) through the disk diffusion method on agar plates [92,93]. Despite its level of standardization, this method requires strict quality controls and high personnel expertise. On the contrary, automated susceptibility testing is possible through automated time-lapse microscopy or volatile organic compound detection within about 7 h [94,95,96,97]. Additionally, microfluidic systems and mass spectrometry can analyse susceptibility profiles through peak intensity and bacterial growth evaluation [98].

All the abovementioned advanced technologies furnish a preliminary phenotypic susceptibility profile, whose advantage is the reporting of MIC values. However, preliminary results should be confirmed through conventional methods to complete the antimicrobial drugs’ spectrum. Interestingly, the rapid phenotypic antibiogram can be forwarded to laboratory personnel and clinicians through phone call or by mailing to “smart reporting” system. This option may significantly reduce TAT for critical patients. Finally, rapid susceptibility testing methods always integrate antimicrobial stewardship programs, which help for managing infection control and antimicrobial treatment in the ICU [99,100]. Ultimately, recent technologies offer the opportunity to generate a concentrated microbial suspension from a positive blood sample within 30 min. The obtained suspension can be used to rapidly obtain identification from MALDI-TOF and susceptibility profile from the colourimetric technique. The mentioned progress could constitute the last frontier in terms of rapid diagnostic methods, resulting in an untimely and complete microbiological report within 24 h from the positivity of the blood sample [101,102,103].

Proteins involved in resistance mechanisms of MDR bacteria may represent an ideal target for lateral flow assays (LFAs). These disposables have been developed to detect resistance enzymes of typical aetiological agents in ICUs, such as *E. coli*, *A. baumannii*, *P. aeruginosa*, *S. aureus*, and *E. faecium*. LFAs demonstrated high sensitivity and specificity rates, while also allowing for the possibility to detect the antigens directly from biological samples [104].

### 4.3. The Potential Role of Whole-Genome Sequencing

ICU critical patients may occasionally require ultimate diagnostic frontiers. Published data documented that the conventional pathogens detection is impossible in about 60% of pneumoniae cases and 30% of sepsis episodes within ICU settings [105,106,107]. This condition often leads to empirical treatments which are not targeted and underestimate harmful resistance profiles. Metagenomics tools provide a unique opportunity to characterize and quantify microorganisms from biological samples, so whole-genome sequencing (WGS) is currently under review as a diagnostic complementary tool. High sensitivity and specificity rates emerged from the completed studies.

Briefly, a nucleic acid extraction is followed by library construction, nucleic acid fragmentation, and amplification processes. The workflow requires 6 h to several days, depending on the applied protocol (long-read or short-read technologies) and personnel training [105]. False negative results are possible due to the impossibility of standardizing extraction and purification protocols directly from biological samples. Furthermore, accurate bioinformatic software and expertise are essential to complete the interpretation process. This concept is related to the extended information quantity that emerges from a sequencing analysis, which can gather data about multiple microorganisms on the same sample (false positive results). Public databases are usually consulted to obtain sequencing data about pathogen identifications, virulence factors, and resistance genes [105,106].

Different studies tested the WGS as an advanced diagnostic technique, comparing its accuracy to gold standards such as culture assays and real-time PCR protocols. Clinical samples like cerebrospinal fluid, respiratory secretions (tracheal aspirate, bronchoalveolar lavage), and blood were included in the experimental protocols. Unfortunately, some genome coverage limitations emerged about low-diffused resistance genes and less common microbial species. Moreover, the lack of infrastructure and sufficient economic support lead to WGS underutilization in ICUs. Data hypothesize a gradual genome sequencing inclusion into ICU diagnostic workflows after implementations in clinical trials, databases, and personnel-specific training about advanced technologies [105,106,107].

Recent publications suggest the WGS as a potential AST method (WGS-AST) due to the capability to furnish accurate resistance predictions. Furthermore, data about the virulence genes may be highlighted through the same technology. Similar information should be integrated with local epidemiological data and known resistance phenotypes. Despite the absence of solid evidence about this application, the WGS may become an interesting challenge in rapidly providing resistance marker detection [107]. Certainly, these conclusions may emphasize the cruciality of performing genomic analysis and document eventual epidemiological changes of MDR clones.

## 5. Antimicrobial Stewardship Programs in Intensive Care Settings

Antimicrobial consumption is one of the most relevant concerns in ICU settings, mostly due to empirical broad-spectrum treatments on critical patients. Such fragile subjects and their severe infections made the antimicrobial stewardship (AMS) programs’ formation essential [108,109]. These programs aim to monitor antimicrobial use and dosage, optimize possible microorganisms’ elimination, and reduce treatment duration, impacting patients’ management and recovery costs. The mentioned aims require a multidisciplinary committee consisting of microbiologists, epidemiologists, pharmacologists, infectious disease specialists, and other targeted clinicians [109]. The ultimate AMS goal is to minimize and contain antimicrobial resistance in critical settings. Statistics document that about 60% of ICU patients come from other hospital wards, claiming the importance of applying the same AMS strategies to all healthcare units. According to the AMS program application, rapid pathogen identification is essential for the early administration of targeted therapies and for avoiding broad-spectrum antibiotic use. Additionally, all the targeted therapies need to be administered depending on the patient’s physiological conditions and the drug’s pharmacological features. Finally, the antimicrobial treatment should be monitored through specific biomarkers and clinical parameters to prevent unnecessary prolonged regimens [108].

The AMS strategies require frequent educational programs for all the involved healthcare personnel with a standardized consensus document about all the aims and protocols. In conclusion, economic and logistic support are vital to plan and reach the AMS goals [108,109]. For instance, clinicians may promptly identify the urgency of an ICU patient’s condition through laboratory biomarkers, imaging, symptoms, and clinical scores. The suspicion of infection should be followed by the request of a fast-track protocol.

Consequently, the laboratory personnel must apply all the rapid technologies to satisfy these requirements. Figure 2 illustrates a suggested fast-track protocol application for ICU patients’ microbiological diagnosis management.

## 6. Conclusions

Intensive care patients represent a concerning healthcare category due to their fragilities and infection predisposition. The clinical microbiologist has the responsibility of providing early and continuous support to clinicians in managing ICU patients. An essential role is to prevent infections by monitoring the patients during their recovery. For this reason, surveillance programs must be encouraged as the prior infection control strategy in the analysed patient setting. Avoiding infections is the key to reducing critical patients’ fatal outcomes. At the same time, once an infection has occurred, it is vital to promptly provide a concrete plan for patient intervention. As a result, rapid identification results and preliminary susceptibility data become the key to succeed in managing the patient and limiting possible extended outbreaks.

All these phases require punctual communication between the clinicians and the laboratory personnel about the patient’s conditions and microbiological evidence. This kind of approach allows the microbiologist to certify the real urgency in rapidly processing a clinical sample instead of applying conventional protocols.

The microbiological report should include all the information that helps clinicians to apply the best clinical practice. In relation to this assumption, a complete report must contain data about identification, phenotypic susceptibility profile, eventual microbial count, resistance markers typing, and indications of the adopted guidelines.

The application of a complete and rapid microbiological protocol is fundamental both for surveillance samples and the specimen collection of a suspected infection. On one hand, a real-time surveillance activation allows for better and faster infection control procedures. On the other hand, a colonization knowledge provides essential information in the case of sudden signs for infection among ICU patients.

The ICU patient always suffers from time-dependent clinical conditions; thus, clinicians should be prepared to provide rapid management and respond to diagnostic requests. On the other hand, clinical microbiologists must satisfy such a critical condition by applying all the available rapid technologies. In conclusion, the ICU patient has inspired technological diagnostic improvements due to the severity of infection and the need for a time-saving diagnosis. In our opinion, the ICU setting could represent a forefront to encourage further studies on advanced diagnostic tools, whose performances will become more essential in the future to rapidly define a microbiological diagnosis.

## Figures and Tables

**Figure 1 antibiotics-13-00238-f001:**
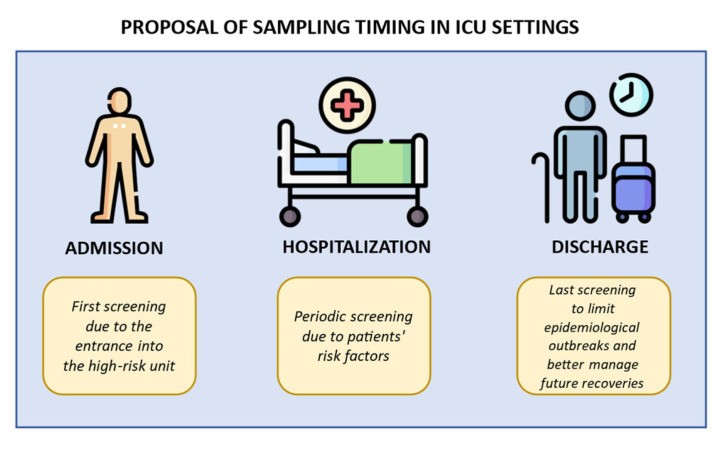
Indications for sampling timing in ICU settings. Created using image database https://www.flaticon.com (accessed on 7 February 2024). The first step is the admission, when a primary screening is performed before future admittance into a high-risk unit. The second step is the hospitalization, when weekly screenings are performed due to patients’ risk factors (catheters implantations, immunosuppression, or others). The final step is the discharge, when a last screening is performed to better manage the epidemiological outbreak and patients’ future recoveries.

**Figure 2 antibiotics-13-00238-f002:**
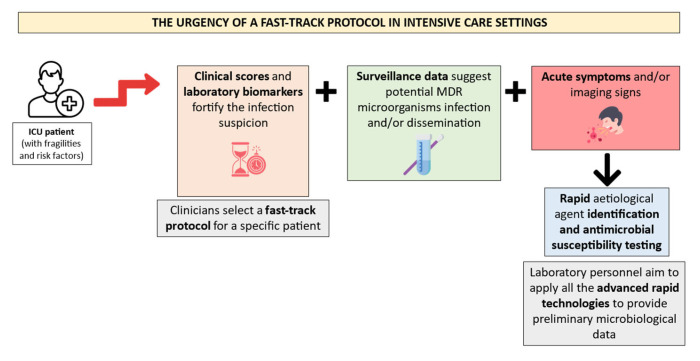
Proposal of a diagnostic workflow organization in ICU patients. Created using image database https://www.flaticon.com (accessed on 14 February 2024). The ICU patients must hold several risk factors and fragilities to require a fast-track protocol after the suspicion of a severe infection. This suspicion is often fortified by clinical scores and laboratory biomarkers, along with MDR colonization data and acute symptoms. Rapid identification and antimicrobial susceptibility testing technologies should be applied by the laboratory personnel in the abovementioned episodes.

**Table 1 antibiotics-13-00238-t001:** Proposal of a surveillance sampling strategy into ICU settings.

	Feaces or Rectal Swab	Perineal Swab	Throat Swab	Nasal Swab	Others
MRSA	Yes ^1^	Yes	Yes	Yes	Yes ^2^
VRE	Yes	Yes	No	No	Yes
MDR *Enterobacterales*	Yes	Yes	Yes	No	Yes
MDR *Acinetobacter baumannii*	Yes	Yes	Yes ^3^	No	Yes ^4^
MDR *Pseudomonas aeruginosa*	Yes	Yes	Yes ^3^	No	Yes ^4^

^1^ These samples are not always requested in intensive care settings, with only some documented data of their usefulness. ^2^ Tracheal aspirate in the case of mechanical ventilation, along with samples from chronic ulcers and urine from catheters may be collected in the case of unconsciousness. ^3^ Sputum or other respiratory samples may be collected along with throat swab. ^4^ Samples from wound exudate may be collected.

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
