# Peer review of "Bacterial Infections in Intensive Care Units: Epidemiological and Microbiological Aspects"

_antibiotics, 2024, doi:10.3390/antibiotics13030238_

Round 1
Reviewer 1 Report
Comments and Suggestions for Authors
The main part of this manuscript covers two aspects: 1) Common pathogens identified in ICU; 2) The conventional and new methods to detect these pathogens. However, the title refers to the role of microbiologist and microbiological reports. The title and the manuscript text do not match at all. This reviewer strongly recommends the modification of this manuscript title. With the existing title, this review article needs and cite and report the topics such as: what is the quality of a well-qualitied microbiologist? What if the microbiologist did it wrong? What are standards of a microbiological report and what are the consequences of a right/wrong reports? While this review is overall happy with the structure and text in this manuscript and appreciate the hard work from the authors (108 references are cited), the authors must come up with a new title, reflecting the information reviewed in this article.
The bacterial name should be italicized. For example, refer to lines 113, 203;
Line 340: 4.1. should be 4.2.?
There are two figures in this review article. The readers would appreciate it if the authors could provide a detailed figure legend so that the authors can follow the information in the figures better.
Author Response
- The main part of this manuscript covers two aspects: 1) Common pathogens identified in ICU; 2) The conventional and new methods to detect these pathogens. However, the title refers to the role of microbiologist and microbiological reports. The title and the manuscript text do not match at all. This reviewer strongly recommends the modification of this manuscript title. With the existing title, this review article needs and cite and report the topics such as: what is the quality of a well-qualitied microbiologist? What if the microbiologist did it wrong? What are standards of a microbiological report and what are the consequences of a right/wrong reports? While this review is overall happy with the structure and text in this manuscript and appreciate the hard work from the authors (108 references are cited), the authors must come up with a new title, reflecting the information reviewed in this article.
Thanks for the observation. The authors decided to modify the title.
- The bacterial name should be italicized. For example, refer to lines 113, 203;
The bacterial species have been italicized.
- Line 340: 4.1. should be 4.2.?
Yes. It has been corrected. - There are two figures in this review article. The readers would appreciate it if the authors could provide a detailed figure legend so that the authors can follow the information in the figures better.
The figures have been enriched with detailed captions.
Reviewer 2 Report
Comments and Suggestions for Authors
The manuscript entitled "Infections in Intensive Care Units: the essential role of the microbiologist and the impact of the microbiological report" aims to attract attention to importance of microbiologists and their reports about the bacterial infections in the ICUs. However, the title and the content of the manuscript are not compatible that much. The majority of the content is about the most common ICU pathogens and their diagnosis methods. The other concerns are:
- Title 2 and 3 are both "Surveillance and diagnostic algorithms" and they do not contain a specific "algoritm".
- It would be better to revise the diagnostic methods. Under the title "4.1. Rapid diagnostic methods", MALDI-TOF, PCR etc were mentioned; however, they are not rapid tests. Mostly, the tests such as lateral-flow assay (LFA) kits are recognized as rapid tests. These kits are not mentioned in the manuscript. Also, Real-time PCR was mentioned as gold standard but this method was not explained.
Comments on the Quality of English Language- The Latin names of species and the gene names should be written italic throughout the text.
- "multi-drug-resistant" should be "multidrug-resistant" throughout the text.
- The abbreviation of " Intensive care units" should be provided (Line 13).
- The full name of "MIC" should be provided (Line 24).
Author Response
- The manuscript entitled "Infections in Intensive Care Units: the essential role of the microbiologist and the impact of the microbiological report" aims to attract attention to importance of microbiologists and their reports about the bacterial infections in the ICUs. However, the title and the content of the manuscript are not compatible that much. The majority of the content is about the most common ICU pathogens and their diagnosis methods.
Thank you for the observation. The authors decided to modify the title. - Title 2 and 3 are both "Surveillance and diagnostic algorithms" and they do not contain a specific "algorithm".
The titles have been revised.
- It would be better to revise the diagnostic methods. Under the title "4.1. Rapid diagnostic methods", MALDI-TOF, PCR etc were mentioned; however, they are not rapid tests. Mostly, the tests such as lateral-flow assay (LFA) kits are recognized as rapid tests. These kits are not mentioned in the manuscript. Also, Real-time PCR was mentioned as gold standard but this method was not explained.
The authors indicated MALDI-TOF and Real-time PCR technologies as rapid methods due to the possibility to perform microbiological assays directly from biological samples. However, a sentence related to LFA has been added.
- The Latin names of species and the gene names should be written italic throughout the text.
They have been corrected. - "multi-drug-resistant" should be "multidrug-resistant" throughout the text.
It has been corrected within all the text. - The abbreviation of " Intensive care units" should be provided (Line 13).
It has been revised. - The full name of "MIC" should be provided (Line 24).
It has been provided.
Reviewer 3 Report
Comments and Suggestions for Authors
The article is well-written and highlights a problem present in almost all intensive care units. For improvement, I would add a chapter on Clostridioides difficile and another on Candida auris.
Author Response
- The article is well-written and highlights a problem present in almost all intensive care units. For improvement, I would add a chapter on Clostridioides difficile and another on Candida auris.
Thank you for the observation. We decided to add a chapter about Clostridioides difficile. However, the present article is focused on bacterial infections, so we decided to not mention fungal species. In our opinion, a future analysis specifically focused only on fungal infections in ICU may be provided in other papers by the authors.
Round 2
Reviewer 2 Report
Comments and Suggestions for Authors
The concerns in the previous version of the manuscripts have been mostly addressed after revision. There are some minor issues related with the writing:
- Since "multidrug-resistant" was abbreviated as "MDR" in line 49, the abbreviation should be used after then.
- The text between the lines 108-116 was repeated in the following paragraph. Also, the "multidrug microorganism" should be "multidrug-resistant microorganism".
- The names of the bacterial species are written italic in the normal text; however, if the text is italic, the species name should not be italic for emphasis. Therefore, the species names should not be italic in the subtitles written italic.
Author Response
The concerns in the previous version of the manuscripts have been mostly addressed after revision. There are some minor issues related with the writing:
- Since "multidrug-resistant" was abbreviated as "MDR" in line 49, the abbreviation should be used after then.
- Sorry for these errors. They have been corrected.
- The text between the lines 108-116 was repeated in the following paragraph. Also, the "multidrug microorganism" should be "multidrug-resistant microorganism".
- We apologize for the repetitions. It must have been a cut/paste error.
- The names of the bacterial species are written italic in the normal text; however, if the text is italic, the species name should not be italic for emphasis. Therefore, the species names should not be italic in the subtitles written italic.
- Thank you for the suggestion. We corrected the font.